# The Quality of Life of Children Facing Life-Limiting Conditions and That of Their Parents in Belgium: A Cross-Sectional Study

**DOI:** 10.3390/children10071167

**Published:** 2023-07-05

**Authors:** Marie Friedel, Isabelle Aujoulat, Bénédicte Brichard, Christine Fonteyne, Marleen Renard, Jean-Marie Degryse

**Affiliations:** 1Department of Life Sciences and Medicine (DLSM), Faculty of Sciences, Technology and Medicine (FSTM), University of Luxembourg, 4365 Esch-sur-Alzette, Luxembourg; 2Institute of Health and Society (IRSS), Université Catholique de Louvain, 1200 Brussels, Belgium; 3Faculty of Public Health, Université Catholique de Louvain, 1200 Brussels, Belgium; isabelle.aujoulat@uclouvain.be; 4Interface Pédiatrique, Department of Paediatric Oncology and Haematology, Cliniques Universitaires St Luc, 1200 Brussels, Belgium; benedicte.brichard@uclouvain.be; 5Globul’home, Hôpital Universitaire des Enfants Reine Fabiola, 1020 Brussels, Belgium; christine.fonteyne@huderf.be; 6Kites, Department of Paediatric Oncology and Haematology, Universitair Ziekenhuis Leuven, 3000 Leuven, Belgium; marleen.renard@uzleuven.be; 7Department of Public Health and Primary Care, Katholieke Universiteit Leuven, 3000 Leuven, Belgium; jan.degryse@kuleuven.be

**Keywords:** Belgium, children’s palliative outcome scale (CPOS-2), life-limiting conditions, outcomes, parents, paediatric palliative care, patient-centred outcome measures, quality of life

## Abstract

Background: Paediatric palliative care (PPC) aims to improve children’s quality of life, but this outcome is rarely measured in clinical care. PPC is provided in Belgium through six transmural paediatric liaison teams (PLTs) ensuring continuity of care for children with life-limiting or life-threatening conditions (LLC/LTC). This study aims to measure the quality of life (QoL) of children with LLC/LTC followed-up by PLTs and the QoL of their parents. Methods: During interviews, an original socio demographic questionnaire, the Children palliative outcome scale—version 2 (CPOS-2), the Fragebogen für Kinder und Jugendliche zur Erfassung der gesundheitsbezogenen Lebensqualität (KINDL) and the Quality of life in life-threatening Illness-Family caregiver (QOLLTI-F) were filled in by PLT members. Statistics were used to investigate significant differences between scores. Results were discussed and interpreted with six PLTs. Results: 73 children aged 1–18 were included in the study. Especially for items focusing on emotional items, children reported their QoL as higher than their parents did. The QoL scores were not significantly associated with the child’s condition’s severity. Conclusions: This study provides, for the first time, an overview of the QoL of children and parents followed-up by PLTs in Belgium.

## 1. Introduction

The main goal of paediatric palliative care (PPC) is to improve quality of life (QoL) for both the child and the family [1,2]. QoL, however, is a complex construct defined by the World Health Assembly as “an individual’s perception of their position in life in the context of the culture and value systems in which they live and in relation to their goals, expectations, standards and concerns” [3].

QoL is a multidimensional construct and is most likely not solely influenced by the burden of disease or the quality of care. “It is important to recognize that experience of care is not the same as outcomes of care. Experiences are likely to be better if outcomes are better, but they relate more closely to how individuals are respected, listened to and heard” [4].

Instruments to measure quality of life in the PPC population are lacking due to many methodological, clinical and ethical challenges [5,6,7].

Outcome measurement research has been considered a research priority in the field of PPC to aid in better responding to patient needs and enhancing individualised care [8,9,10]. In recent years, patient-centred outcome measures (PCOM), which encompass both patient-reported and proxy-reported measures, have been further developed. An outcome measurement instrument evaluates ‘change in health status’ as a consequence of health care or interventions [11]. PCOM might prevent cognitive bias by reducing the risk of relying on perceptions from health-care professionals towards children’s or parents’ quality of life.

A meta-summary was compiled to identify the meaningful outcomes of PPC. As a result, eight themes were listed: the relationship with professional caregivers, pain and its management, “living beyond pain”, the relationship between paediatric patients and their families, children’s views on their treatment and service provision, meanings children give to their end-of-life situation, consequences of clinical decisions and the relationships among children in paediatric palliative care and their peers [12].

These findings were confirmed by a systematic review conducted by Namisango et al. which looked at the meaningful domains of PPC [13]. Five domains reflected priority concerns: physical (e.g., symptoms), psychological (e.g., worries), psychosocial (e.g., relationships), existential (e.g., existential loss) and others (e.g., information access). The results showed that children’s perspectives were not systematically researched.

Despite the methodological challenges to developing an outcome measurement instrument, a promising tool developed by the African Palliative Care Association, called the APCA children’s palliative outcome scale, was developed in an African context of care [8,14]. Based on the APCA palliative outcome scale for adults [15], it combines a self- and proxy-report. The CPOS is a multidimensional patient-centred outcome measure with five-point Likert-scale response options. It contains 12 items exploring physical and psychosocial elements. Seven items are related to children’s outcomes, which may be rated by the children themselves (self-report) and rated by their parents (proxy-report), and five items focus on parents’ outcomes.

In Belgium, paediatric palliative care is provided through paediatric liaison teams (PLTs) attached to university hospitals. They offer curative, palliative and liaison care as a mobile team available 24/7. Services provided in a family-centred approach to ensure continuity of care are free of charge for users. Previously published studies have shown that more than 700 children aged 0–18 years are monitored on an annual basis by teams in Belgium [16]. Their access to children facing complex chronic conditions in the Brussels region, however, has been limited [17].

A previously published pilot study on the face/content validity, acceptability and feasibility of the original children’s palliative outcome scale was conducted among 14 children, 19 parents and 9 representatives of PLTs [18]. During this pilot study, the original African Palliative Care Association APCA-CPOS [8] was further developed by adding items to the scale which were found to be more meaningful for children. In fact, children were invited to self-elicit domains of quality of life (QoL) via an instrument called the Scheduled Evaluation of Quality-of Life direct weighted (SEIQoL-DW) [19], resulting in a CPOS version 2 (CPOS-2) which includes 22 items.

After the pilot test, a larger field study was then conducted with the adapted CPOS-2 among children and parents followed-up by PLTs.

This paper presents the results of this field study, providing for the first time an overview of the sociodemographic characteristics and QoL of children facing LLC/LTC and that of their parents, followed-up by six PLTs in Belgium during a one-year period.

We will first present the characteristics of the population and then focus on three research questions:How do the scores on the new CPOS-2 and the often used Quality of life in life-threatening illness-family carer version 2 (QOLLTI-F) relate to scores observed in other populations?How does the appreciation for the well-being of parents and children by the responsible paediatricians correlate with quality of life as measured by the QOLLTI-F (parents) and the CPOS-2 (children’s self- and parents’ proxy-report)?Which characteristics and/or background variables are associated with the scores on the QOLLTI and/or the CPOS-2?

## 2. Materials and Methods

This study was conducted in Belgium from 1 February 2019 to 1 March 2020. Six PLTs (two in the region of Flanders, two in the region of Brussels and two in the region of Wallonia) participated in this study. All study material was available in 2 languages, French and Dutch.

This research has been performed in accordance with the Declaration of Helsinki. Informed assent was obtained from all included children and informed consent was obtained from their parents and/or legal guardian. Children and parents cared for by paediatric liaison teams were invited to participate in the interviews. Specific age-appropriate information and assent letters were provided, and each child/adolescent with cognitive capacities was invited to read and to affirm it.

All children/adolescents and their parents followed-up by PLT and responding to the inclusion criteria were invited by the teams to participate in the study. Families were invited to take part in an interview conducted by members of a PLT. Purposive sampling of all children and adolescents cared for by one of the 6 PLTs in Belgium was carried out with the following inclusion criteria: >one year old; not at an imminent end-of-life stage (last 7 days of life); parents able to understand French or Dutch; consent obtained from parents and assent from children. Children were excluded if they were in their last days of life, if parents were not able to understand French or Dutch or if they were <1 year old. For each family, quantitative data were collected through several questionnaires, which were printed on carbon sheets to keep the original within the PLT and a copy provided to the research team. A code was allocated to each family to respect their confidentiality.

Each PLT completed a document indicating the number and characteristics of non-invited children/parents, presenting exclusion criteria (and rationale) and invited children/parents to the study who refused to participate (and rationale).

A member of the PLT completed the following questionnaires during an interview with each participating family:An original 35-item sociodemographic and medical questionnaire including 5 questions from the Paediatric Palliative Screening Scale [20,21] evaluating a child’s life expectancy, the impact of the disease and the impact of treatment on children’s daily lives and the level of the child’s, parents’ and siblings’ suffering/distress as perceived by the physician and rated on a 3-point Likert scale. This 35-item questionnaire was developed by our research team and previously discussed with representatives of three different paediatric liaison teams.The self- and proxy-report 22-item CPOS-2 [8,18] evaluates children’s and parents’ quality of life. Each item is scored on a 5-point Likert scale ranging from 0 to 5. The items are worded in such a way that a higher item score implies a poorer quality of life or, more precisely, an elevated burden. The values provided by 12 out of the 22 items are reversed to explore the construct in the same direction. The total score ranges between 0 and 60 for Part A and between 0 and 50 for Part B. These scores are calculated in percentage scores for ease of interpretation. The scale contains reflective items and formative items and is thus considered a hybrid structure. A cross-cultural translation of the CPOS in French was made according to the guidance of Antunes et al. [22] and De Vet et al. [23], and its face/content validity, acceptability and feasibility were reported in a previous paper [18]. A translation of the CPOS-2 into Dutch was carried out by our research team in close collaboration with representatives of the paediatric liaison teams based in UZ Leuven and UZ Gent. Agreement to use the CPOS-2 in our study was obtained from the tool designers.The generic 35-item Fragebogen für Kinder und Jugendliche zur Erfassung der gesundheitzbezogenen Lebensqualität (KINDL) [24], customised for three age categories (3–6 y, 7–13 y, 14–17 y), evaluates children’s health-related quality of life through child (self) and parental (proxy) reports. The original KINDL is a validated scale to measure quality of life. The higher the score, the higher the quality of life. We reversed the scoring of the items and used it as a scale to measure the impact of the disease on the quality of life (the burden). Therefore, high scores represent a high burden and vice versa. The scale was developed to measure quality of life in healthy as well as in ill children. The available translated versions in Dutch and French were used for our study. Agreement to use the KINDL in our study was obtained from the tool designers.The validated parental self-report 17-item Quality of life in life-threatening illness-family carer version 2 (QOLLTI-F) [25] assesses parental quality of life. The QOLLTI-F originates from qualitative interviews exploring family burdens as well as positive experiences in the caregivers’ situation, which are included in the questionnaire. The QOLLTI-F includes seven subscales assessing different domains: environment, patient condition, the caregiver’s own state, caregiver’s outlook, quality of care, relationships and financial worries. All items have a possible range from 0 to 10, with 0 indicating the worst situation and 10 the best. A high total score indicates a good situation (a high quality of life or a low burden). Five items are transposed prior to calculating subscales and total scores. All subscale scores are calculated by taking the mean of the items comprising that subscale. The QOLLTI-F total score is the mean of the subscale scores. A French version of the QOLLTI-F was already available. A Dutch version was produced by our research team in close cooperation with paediatric liaison team members from UZ Leuven and UZ Gent for linguistic and semantic soundness. Agreement to use the QOLLTI-F in our study was obtained from the tool designers.

Descriptive statistics were conducted on the sociodemographic data, the CPOS QoL scores and the QOLLTI-F scores. The association between clinicians’ judgement of quality of life and the scores on the QOLLTI-F and CPOS-2 was explored using Spearman’s correlation coefficients and boxplots. We used the highest quartile value of the QOLLTI-F score (quality of life parents) and the highest quartile value of the CPOS-2 score (impact on the quality of life of the family) to define contrasting groups and compare the characteristics of families with higher quartile values with the others. Pearson’s chi square test and Fisher’s exact test were used to investigate significant differences. Analysis was performed with SPSS 26. (SPSS Inc., Chicago, IL, USA).

The results of the descriptive statistics were presented to each of the six paediatric liaison teams in June 2020 during four different virtual meetings, and their comments and questions enriched the interpretation of the data.

## 3. Results

### 3.1. Sample

A total of 73 children/adolescents (1–18 years) and their families were included in the study. The primary reason for the exclusion of families in the study, despite meeting the inclusion criteria, was lack of time to commit, as reported by PLTs. Only five families refused to participate. Table 1 shows the characteristics of the included children. A homogeneous proportion of girls and boys was found in our sample. A high proportion of children were <6 years old (65%), while few teenagers were included (10%). Regarding the pathology, most of the children included in the study had either a neurological or a metabolic/genetic disease (61%), and few (25%) faced an oncologic disease. More than half (55%) of the children received artificial nutrition, and one-fifth needed respiratory support, such as oxygen administration, continuous positive airway pressure or non-invasive respiratory support. Only 19 (26%) of all included children had verbal capacities. Thirty children (41%) received psychological support. At the time of interview, more than half (53%) of the children included had been monitored by a PLT for more than one year.

We found that our sample was representative of age ranges and gender but not of categories of diseases. In fact, onco-haematological diseases were underrepresented in our sample compared with the population of children monitored by paediatric liaison teams. 

Regarding socioeconomic elements, 76% of the included children had parents living as a couple, whereas 18% of the children lived in a single-parent family. Forty-five percent (*n* = 33) of the included parents (*n* = 73) reported having been obliged to completely dismiss their professional work to care for their child at home. Of those parents (*n* = 33), mothers were predominant (*n* = 25, 76%) compared with fathers (*n* = 2, 6%). For six families (18%), both parents stopped professional work completely. Eighteen percent (*n* = 13) of families included in the study reported benefitting from social or financial assistance. Furthermore, only 39 parents (53%) declared receiving psychological support.

Focusing on quality-of-life scores, Table 2 shows the mean scores self-reported by children on their quality of life as measured by the CPOS-2 and the KINDL. Parents’ proxy reports through CPOS-2 and KINDL are also indicated. Finally, parents’ perspectives on their own quality of life as measured by the CPOS-2 Part B and the QOOLLTI-F are also included.

We found a significant correlation between scores on Part A of the CPOS-2, Part A (measuring children’s QoL) as reported by children and that as reported by parents (observed correlation coefficient 0.55, and corrected correlation coefficient 0.79). We also looked at each item in detail by computing correlations between children’s and parents’ scores. We found that for questions 1, 2, 6, 8, 10 and 12 of the CPOS-2, the correlations were statistically significant, whereas for questions 3, 4, 5, 9 and 11, all linked to emotional status, the scores between children and parents differed considerably.

Q3 Is there anything about food that has been bothering you?

Q4 Can you tell me if you have been sad?

Q5 Can you tell me if you have been happy?

Q9 When something bothers you, can you talk to someone about it?

Q11 If you had a magic wand, is there something you would like to change in your family?

Regarding their quality of life, we found that parents had a relatively low score of quality of life as rated on the QOLLTI-F (mean of 64.11) and on the CPOS-2 Part B (mean of 59.15). The scores were the lowest in the two subscales of the QOLLTI-F called Patient State and Carer’s own state (Figure 1). These two subscales correspond to the two following questions:

“During the last 2 days, the condition of my child, whom I care for, has afflicted me.” and

“During the last 2 days, the level of control I have on my life has been a problem”.

### 3.2. Correlation of QoL Scores Found in Our Study with Other Populations

Our first research question aims to compare QoL scores in our study with those found in other populations. Table 3 offers an overview of scores produced by KINDL and QOLLTI-F in other populations [26,27,28,29,30,31]. We found that the means of the children’s KINDL scores were close to those found in our study (approximately 70%). The norm in healthy children is 76.8% [26,27]. Two other studies, one in children with diabetes [28] and another among children with congenital heart disease [29], found that the mean QoL scores were higher than those of healthy children of the same age reference group.

Parental QoL scores found in our study, as measured by the CPOS-2 and the QOLLTI-F, were comparable to the scores found in two other studies conducted among parents who had a child receiving palliative care [30,31]. Those studies pointed to the impact of financial, emotional and physical dimensions on parents’ QoL.

### 3.3. Assessment of Child’s, Parents’ and Siblings’ Suffering by the Paediatrician

Our second research question concerns the correlation between the assessment of the level of suffering of parents and children by the responsible paediatrician, with the quality of life as measured by the QOLLTI (parents QoL) and the CPOS (children’s QoL).

We found that for 45 children (62%), their specialist paediatrician estimated that a child’s life expectancy was completely unpredictable (Table 1). Moreover, they perceived that for 67% of the children, illness may have a very high impact on their daily activity, and for 34% of them, medical treatment would have a very high impact on the children’s quality of life.

Paediatricians rated the level of parental suffering as very high for 39 parents (57%), whereas the child’s suffering was estimated as being very high for 28% of them. For 13% of the included children, paediatricians found the level of the child’s suffering was difficult to estimate. Siblings’ suffering was found to be difficult to estimate in 32% of cases (Table 4).

We conducted correlation studies to compare children’s and parents’ perceptions of their QoL (as measured by the CPOS-2 and the QOLLTI-F) with paediatricians’ perceptions of their level of suffering. As shown in Table 5 and Figure 2, we found that paediatricians frequently found it difficult to assess the level of distress/suffering in siblings (and to a lesser extent in children). However, when they make an assessment of children’s and parents’ suffering, their judgement correlates well with scores of the QOLLTI-F as well as of the CPOS (Part B for the parents and Part A for the children). It should be noted that a high score on the CPOS-2 indicates a high burden or impact on QoL, whereas a high score on the QOLLTI-F shows a high quality of life (low burden or impact of the disease).

### 3.4. Factors Associated with QOL as Measured by the QOLLTI and the CPOS-2

Finally, as a third focus, we explored which characteristics and background variables were associated with the highest quartile scores on the QOLLTI-F (high score = high quality of life) and/or the CPOS-2 (high score = higher impact of the disease or lower quality of life). As seen in Table 6, we found no association with the age or sex of the child; with the type of disease the child was suffering from; or with the length of the follow-up by the PPC teams. Four parental characteristics were also investigated: socioeconomic status, level of education, ability to cope with the situation (as evaluated by the PPC nurse) and the perceived impact on daily life, with no significant associations found (not reported in the table).

The most striking finding was that no association was found between any indices describing the degree of disability of the child and the quality of life of parents, as measured in our sample by the QOLLTI-F, and in the family as measured by the CPOS. The only two factors associated with a high QOLLTI-F score were the estimation by the paediatrician of the child’s and parents’ level of suffering.

A high CPOS-2 score was associated with the paediatrician’s perception of parents’ and children’s level of suffering and the need for specific equipment, particularly the presence of a nasogastric feeding tube and/or the need for oxygen therapy.

These results, however, must be evaluated with caution because of the small size of our group (*n* = 73).

## 4. Discussion

This study provides, for the first time, an overview of the QoL of families followed-up by a PLT in Belgium in 2019. A sample of 73 children aged 1–18 years and their parents were included. It shows a slight discrepancy between children’s and parents’ scores on children’s QoL, as measured by the CPOS-2 and the KINDL. Especially for items focusing on emotional items, the children included in our study self-reported their QoL as higher than their parents did. Children’s QoL, as rated by the CPOS-2 and the KINDL, demonstrated a relatively high QoL (mean of 72/100) compared with parents’ QoL as measured by parents themselves on the QOLLTI-F (mean of 64/100) and the CPOS-2 Part B (59/100).

With reference to other studies [26,27,28,29], we found that the mean child’s QoL scores, as rated by the CPOS and the KINDL, were similar. Those studies confirmed our results suggesting that the child’s health-related QoL, as measured by KINDL, is not significantly associated with the severity of the child’s condition or their impaired activities of daily life. Several hypotheses can emerge from these findings. Children can adapt over time to their disease and condition, showing increased resilience. A second reason can be linked to response shift and a third to the provision of high medical care offered to families, helping them to feel empowered and supported. Those studies, however, also indicated that QoL might be highly influenced by relations with peers, especially for adolescents, and not solely by the burden of the disease. Furthermore, a study conducted among children and adolescents with cerebral palsy who self-reported their quality of life, as measured by the instrument Kidscreen, found that 8- to 12-year-old children had similar QoL scores to those in the general population, whereas adolescents with cerebral palsy (13–17 years) had significantly lower QoL on only one domain (social support and peers) [32].

In our study, we found that paediatricians could not estimate the life expectancy for 45 of the children included (62%). This contrasts with the surprise question (“Would you be surprised if your patient would die in the next 6–12 months?”) often used as a valid criterion in adult palliative care. In Belgium, the palliative care indicators tool for adults, PICT, was officially introduced as a referral criterion to palliative care services and includes the surprise question [33]. For a paediatric palliative care population, one single prospective cohort study found that the surprise question on children’s life expectancy was a highly sensitive prognostic tool for identifying children who are in the last 3 to 12 months of life [34]. In another qualitative study conducted among 10 Belgian specialist paediatricians in a single university hospital, not being able to attend school, an intuitive perception of the family’s suffering, the ability to cope and the need to assist the child technically with medical equipment at home were the most-reported criteria by hospital paediatricians for referral to a paediatric liaison team [35]. This is consistent with the findings of this field study, in which the presence of medical equipment and the paediatrician’s perceptions of a family’s level of suffering were statistically significantly associated with a high CPOS-2 score.

In our study, we found that parents often wanted to talk about prior events they had experienced rather than limiting themselves to the last two days (as requested by the CPOS, KINDL and QOLLTI-F questionnaires). This suggests that some experiences linked to the announcement of the diagnosis or the care pathway might be enduring or even traumatic. This is in line with the results of two studies looking at parental suffering, measured by the degree of posttraumatic stress disorder (PTSD) when their child faced a serious illness. Indeed, 10% of mothers and 18% of fathers showed full PTSD, even five years after the child’s end of cancer treatment [36,37]. Quality of life might be associated with a capacity of resilience, while parental resilience may differ from adolescent resilience. A recent study among cancer patients and their mothers used the Connor-Davidson Resilience Scale (CD-RISC-10), which is a 10-item measure of resilience, personal problem solving and approaches to adversity, and showed that “higher adolescent and young adult (AYA) distress predicted better maternal resilience, whereas higher maternal distress predicted worse AYA resilience” [38]. These results suggest that the process of resilience varies between mothers and adolescents.

The discrepancy between self- (child) reports and proxy- (parent) reports of children’s health-related QoL found in our study echoes the results of other studies [39,40,41,42,43]. These different perceptions were found particularly in psychosocial domains of health-related QoL, such as emotional functioning, and these differences were age-related [44,45]. Although some authors [46] suggest that a single proxy report (by parents) would be enough for rating children’s pain intensity, for example, other authors recommend always requesting child and parental reports to acknowledge the different perceptions of each and to foster the acceptance of the psycho-social dynamics of the child–parent dyad made of loyalty, trust and interdependence [9,43,44,47,48].

On the other hand, health-care professionals and parents in our study appeared to share similar perceptions of the parents’ QoL, which in clinical contexts is not always the case. If clinical teams perceive a family’s quality of life as low, cognitive bias might jeopardise the shared decision-making process. To prevent this, Carnevale et al. suggest recognising first “the children as active agents, that is, persons who have interests and capacities to participate in discussions and decisions that affect them and other people” [49]. They also promote an “empathic attunement”, which he describes as an attempt to “sense the emotional perspective of the other but also implies a stunning to grasp the person’s understanding of the situation to the greatest possible extent” [50]. In brief, using patient-centred outcome measurement instruments such as the CPOS-2 might help health-care professionals objectify their representations, and sometimes misconceptions, on children’s and parents’ QoL. On a long-term basis, when used several times with the same family, the CPOS-2 could document the impact of paediatric palliative care on children’s and parents’ quality of life. We do, however, acknowledge that an outcome measurement instrument might not be able to fully capture one’s subjective quality of life, reflected in a score, but might be used as a tool to address sensitive issues, discover unmet needs and facilitate the shared decision-making process. 

It is not yet clear whether paediatric palliative care outcomes overlap with quality of life. We assume that assessing quality of life in children facing life-limiting conditions is a way to assess meaningful outcomes of PPC, which are not limited to the burden of disease. Nonetheless, we are aware that some factors affecting children’s or parents’ quality of life may not be managed or controlled by PPC health-care teams, such as relations with peers and friends. Furthermore, we acknowledge that an outcome measurement instrument will probably never be able to assess quality of life in a perfectly reliable way.

The relatively small sample size included in our study might be a limitation for the generalisation of our findings. It represents 24% (*n* = 73/309) of the whole population of children (with the same inclusion criteria) cared for by paediatric liaison teams in Belgium, as reported by them in a timeframe of 12 months. The instruments used in our study did not allow for self-reports of QoL by nonverbal children (only 19 children included had verbal capacities, see Table 1). This limitation has already been reported in other studies. To overcome this barrier, suggestions include relying purely on parental or professional proxies focusing on core outcomes identified for children with severe neurological impairment [51], using pictograms, drawings [52] or communication boards, electronic touch pads or adapted devices [48].

Further studies with larger sample sizes, especially for children with verbal capacities, are warranted to better document their perspectives on quality of life and identify potential subgroups.

## 5. Conclusions

This study provides, for the first time, an overview of the quality of life of families followed up by a paediatric liaison team in Belgium in 2019, as assessed by the children’s palliative outcome scale-2. Quality of life scores do not seem to be associated with the severity of a child’s disease.

## Figures and Tables

**Figure 1 children-10-01167-f001:**
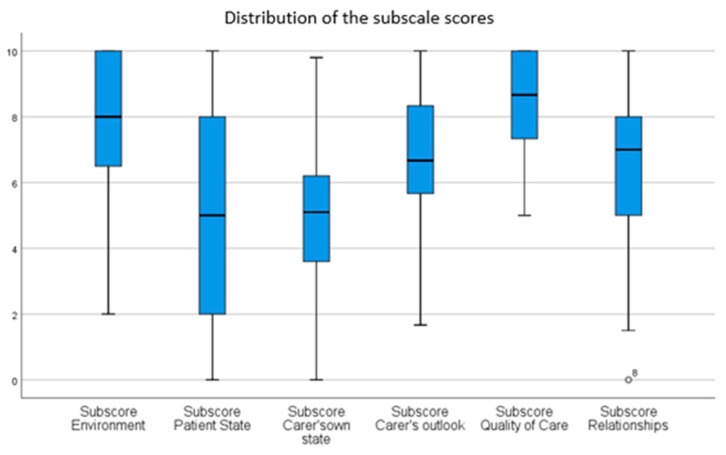
Distribution of the subscale QoL scores as measured by the QOLLTI-F.

**Figure 2 children-10-01167-f002:**
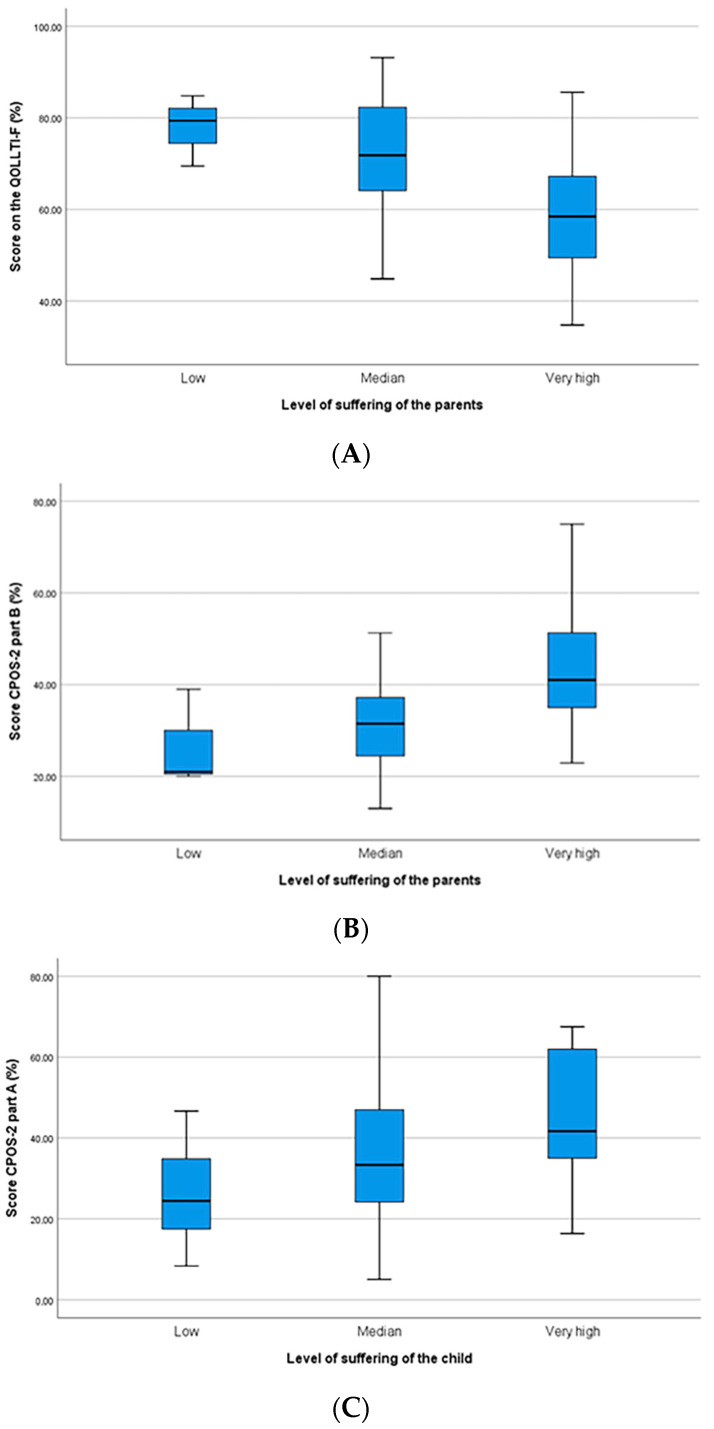
(**A**) Level of suffering in parents as assessed by the paediatrician in relation to the QOLLTI-F scores; (**B**) level of suffering in parents as assessed by the paediatrician in relation to the scores on Part B of the CPOS-2; (**C**) level of suffering in children as assessed by the paediatrician in relation to the scores on Part A of the CPOS-2.

**Table 1 children-10-01167-t001:** Characteristics of children and adolescents included in the field study (total *n* = 73).

Characteristics of Children	*N* (%)
**Gender**	
Total	73 (100%)
Female	37 (50%)
Male	36 (50%)
**Age**	
Total	73 (100%)
1–2 years	18 (25%)
3–6 years	25 (35%)
7–13 years	22 (30%)
14–17 years	7 (10%)
**Disease**	
Total	73 (100%)
Neurology	28 (38%)
Onco-haematology	18 (25%)
Metabolic/genetic	17 (23%)
Neonatology	7 (10%)
Cardiology	3 (4%)
**Verbal capacity**	
(Child is able to understand and respond to questions as judged by parents)	19 (26%)
**Medical equipment**	
Artificial nutrition	
Gastrostomy	32 (44%)
Naso-gastric feeding tube	8 (11%)
**Respiratory assistance**	
Oxygen therapy	7 (10%)
Non-invasive ventilation	6 (8%)
Tracheotomy	3 (4%)
**Intravenous access**	
Port-a-cath/Broviac	5 (7%)
**Length of follow-up by a PLT at time of interview**	
Total	73 (100%)
0–1 month	0 (0%)
1–3 months	4 (5.5%)
3–6 months	7 (9.9%)
6–12 months	11 (15.5%)
1–2 years	11 (15.5%)
2–3 years	9 (12.7%)
>3 years	18 (25.4%)
Missing data	11 (15.5%)
**Child’s life expectancy as assessed by the paediatrician**	
Total	73 (100%)
1–6 months	2 (2.7%)
7–11 months	1 (1.3%)
1–2 years	6 (8.2%)
3–5 years	7 (9.5%)
>5 years	7 (9.5%)
Totally unpredictable	45 (62%)
Missing data	5 (6.8%)

**Table 2 children-10-01167-t002:** Characteristics of children’s and parents’ QOL scores as measured by the CPOS-2, the KINDL and the QOLLTI-F questionnaires.

	Number	Converted to Mean % and Standard Deviation
Children’s QOL		
CPOS-2 reversed score		
Self-report Part A	22	73.19% (15.91)
Proxy-report Part A	70	65.15% (16.25)
KINDL reversed score		
Self-report	23	70.97% (15.31)
Proxy-report	49	58.91% (11.47)
Parental QOL		
CPOS-2 reversed score		
Proxy-report Part B	73	59.15% (12.89)
QOLLTI-F	71	64.11% (13.34)

CPOS-2: Children palliative outcome scale version 2. KINDL: Fragebogen für Kinder und Jugendliche zur Erfassung der gesundheitzbezogenen Lebensqualität. QOLLTI-F: Quality of life in life-limiting illness-family carer. The CPOS-2 and KINDL scores were reversed to align with the QOLLTI-F scores and reflect the quality of life (highest QoL = 100%).

**Table 3 children-10-01167-t003:** Comparison of QoL scores among different studies using the KINDL and the QOLLTI-F.

KINDL	N (Age)	Disease	Results	Mean Score (SD)
Khair et al. (2012), UK [26]	84 (6–17 y)	Haemophilia	The highest impairments (KINDL) in the 8- to 12-year-old group were in the dimension “school” (55.01 ± 17.2) and self-esteem (59.5 ± 17.1), whereas scores for 6- to 7-year-olds were much higher for these dimensions (75.0 ± 31.0 and 75.0 ± 23.1, respectively).	KINDL TOTAL scores 6–7 y: 77.61 (14.2) 8–10 y: 70.40 (8.9) 13–17 y: 70.38 (12.3)
Hövels-Gürich et al. (2007), Germany [29]	40 (5–12 y)	Congenital heart disease	Children 5 to 7 years old reported better QoL (total score) than the same age reference group. Self-reported QoL scores for 8- to 12-year-olds did not differ from those in control subjects in any domain.	Full text not available
Müller-Godeffroy et al. (2008), Germany [27]	50 (6–16 y)	Spina bifida	Children with spina bifida (8–12 y) reported lower HRQoL in all dimensions (“emotional,” “self-esteem” and “friends”) and total score (medium to large effect sizes). Adolescents reported lower scores on peer relations. Most medical parameters as well as limitations in ADL were not significantly associated with HRQoL. Our findings confirm the results of studies which dispute a linear inverse association between condition severity and HRQoL.	8–12 y: 69.6 (95% IC 57.8–67.1) 13–16 y: 69.7 (95% IC 59.6–74.8)
Wee et al. (2005) [28]	30 (mean age: 10.7 ± 1.35 years	Diabetes mellitus	Overall, children with DM reported better HRQoL than healthy children. Although this appeared counterintuitive, several explanations are possible: (1) the development of resilience to the disease over time, (2) our subjects are well-managed, (3) response shift, (4) the provision of high-quality medical care, (5) compared with normal children, diabetic subjects and their families pay greater attention to health issues.	The reliability coefficients were (overall, scales): KINDL-Kid DM (0.79, 0.44–0.65), KINDL-Kid Healthy (0.71,0.60–0.80), KINDL-Kiddo DM (0.77, 0.37–0.74) and KINDL-Kiddo Healthy (0.84, 0.21–0.79)
QOLLTI-F				
Groh (2013), Germany [30]	40 parents	Various life-limiting conditions		QOLLTI-F total score before intervention median 5.8 (IQR: 1) After intervention 7.1 (IQR: 1.3) < 0.001
Bradford (2012), Australia [31]	10 parents	Various life-limiting conditions	Two domains of caregiver quality-of-life require further study: their finances and their emotional and physical state	QOLLTI-F total scores Mean IG: 5.4–6.2 Mean CG: 6.6–7

**Table 4 children-10-01167-t004:** Levels of child’s, parents’ and siblings’ suffering as perceived by the paediatrician (total of 73 children, 100%).

	Difficult to Assess	Low	Medium	Very High	Missing Values
Child’s level of suffering	12.3%	20.5%	34.2%	26.0%	6.8%
Parents’ level of suffering	6.8%	4.1%	30.1%	53.4%	5.5%
Siblings’ level of suffering	24.7%	6.8%	31.5%	15.1%	21.9%

**Table 5 children-10-01167-t005:** Correlation of paediatrician’s assessment of parental suffering with the CPOS-2 and QOLLTI-F scores.

	Level of Suffering Parents	CPOS-2 Score (%) (Part B)	QOLLTI-F Score (%)
Level of suffering Parents		0.47 **	−0.50 **
CPOS-2 score (%) (Part B)	*0.61* °		−0.54 **
QOLLTI-F score (%)	−*0.56* °	−*0.78* *	

Spearman’s correlation coefficients; ** correlation is significant at the 0.01 level (2-tailed); ° coefficients corrected for attenuation. Italic and * correlations corrected for attenuation.

**Table 6 children-10-01167-t006:** Association between baseline characteristics and background variables and highest quartile QOLLTI-F and CPOS-2 scores.

	QOLLTI-F Score, Highest Quartile (*n* = 17)	QOLLTI-F Score, Rest of Quartiles (*n* = 56)	*p* Value	CPOS-2 Score, Highest Quartile (*n* = 21)	CPOS 2 Score, Rest of Quartiles (*n* = 52)	*p* Value
Age			0.55 ^a^			0.46 ^a^
1–2 y	2	16		6	12	
3–6 y	6	20		6	20	
7–13 y	7	15		7	15	
14–17 y	2	4		1	5	
>17 y	0	1		1	0	
**Sex Male (*n*, %)**	7 (43%)	29 (51%)	0.31 ^b^	10 (47.5%)	26 (50%)	0.53 ^b^
Type of pathology			0.10 ^a^			0.84 ^a^
A = Neurological	8	15		6	17	
B = Onco-haematological	1	17		4	14	
C = Metabolic/genetic	6	18		8	16	
D = Neonatal	0	3		1	2	
E = Cardiac	1	0		0	1	
Disability indices						
Verbal capacity			0.15 ^a^			0.58 ^a^
A = Normal	5	18		4	19	
B = A few words, limited vocabulary	0	5		1	4	
C = Vocalisation, sounds, screams	1	12		4	9	
D = No verbal communication at all	6	16		9	13	
Cognitive capacity			0.32 ^a^			0.17 ^a^
A = Normal	5	22		5	22	
B = Limited (limited verbal interaction)	1	5		0	6	
C = Smiles, facial expressions, produces sounds	1	9		4	6	
D = Severely limited, severe developmental disability	7	16		9	14	
Mobility			0.65 ^a^			0.26 ^a^
A = Normal mobility in accordance with age (sitting, walking, etc.)	3	15		2	16	
B = Sits, walks with assistance, axial hypotonia	1	9		3	7	
C = Wheelchair or another device needed	6	12		5	13	
D = Bedridden, no spontaneous mobilisation	5	15		9	11	
Ability to eat			0.26 ^a^			0.46 ^a^
A = Normal	4	17		3	18	
B = Eats with help	3	6		2	7	
C = Naso-gastric tube	6	8		4	10	
D = Gastrostomy	3	19		9	13	
Duration of follow-up			0.73 ^a^			0.69 ^a^
0–1 months	1	3		1	3	
1–3 months	0	7		3	4	
3–6 months	2	9		3	8	
6–12 months	2	9		5	6	
1–2 years	3	6		1	8	
2–3 years	5	13		5	13	

^a^ *p* value based on Pearson chi-square test; ^b^ *p* value based on Fisher exact test.

## Data Availability

The datasets generated and/or analysed during the current study are not publicly available, but are available from the corresponding author on reasonable request.

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
