# Peer review of "The Quality of Life of Children Facing Life-Limiting Conditions and That of Their Parents in Belgium: A Cross-Sectional Study"

_children, 2023, doi:10.3390/children10071167_

Round 1

Reviewer 1 Report

This study provides an overview of the quality of life of families followed by a PLT in Belgium in 2019. Quality of life is the main goal of pediatric palliative care. For this reason, this work is of great importance.

I want to congratulate the authors for this topic. too few doctors are concerned about the quality of life, in the context in which it is desired to extend it in modern medicine.

The article is well written, the introduction presents the problem in detail. The abstract is rather long, 284 paragraphs instead of 200.

The material and method chapter, as well as the results section, are well written. The discussion section could be more developed. I agree that the small sample size included in your study could be a limitation to the generalizability of the findings.

However, there are many studies that have a similar group of patients and that could have been discussed comparatively.

Author Response

Dear reviewer, 

Kind regards, 

Marie Friedel

Reviewer 2 Report

Thank you for giving me the chance to review this work, which targets an important topic for an important and vulnerable population. Authors used a cross sectional survey to assess the quality of life of children facing life-limiting conditions and of their parents.

The study is promising but needs some revisions. In doing so, plz consider the following:

(1)  Lines 150-151: For each of the families who participated in the study, a member of the PLT completed the following questionnaires, one for each interview conducted with the families”. This statement is confusing. Plz rewrite to clarify the last phrase “one of what for each interview”?

(2)  The measures used are also confusing. Plz clarify the purpose of each measure. So both CPOS and KINDL measure children’s qol? Also parents qol are conceptualized and operationalized as different from parental qol right?

(3)  Please clearly state who filled each measure.

(4)  Age categories need a rational. Why a child aged 7 is expected to be similar to another aged 13?

(5)  Please clarify how the data were collected across different age groups.

(6)  The entire data collection protocol needs to be detailed (when, where, and how)

(7)  How were missing data managed?

(8)  I don’t see effect sizes. I think there is an over emphasis on statistical significance without consideration of clinical significance of the findings.

(9)  Do we have measures psychometrics?

(10)                In discussing the discrepancy between self- (child) reports and proxy- (parent) reports of children’s health-related QoL, authors cite previous similar results, but the finding is not discussed.

(11)                Please discuss how the heterogeneity of the sample might limit the findings.

(12)                To what extent authors can really link qol outcomes to the care provided to participants and their parents?

None

Author Response

(The authors gave the same response as above.)

Round 2

Reviewer 1 Report

Dear authors,

I appreciated the changes made according to the recommendations.

My only puzzlement is why there is an identical article, published in 2022, by the same team of authors? Can you explain me? Maybe I misunderstood something...

I have attached a print screen and DOI of this article below.

DOI: https://doi.org/10.21203/rs.3.rs-1478935/v1

Thank you

Reviewer 2 Report

I have carefully reviewed the revised version of the article and I am pleased to see that the authors have adequately addressed all the comments and suggestions raised during the previous review. They have made substantial improvements and addressed the concerns effectively. Therefore, I have no further comments or recommendations at this time.

Note to authors: In correlation analysis, the effect size typically refers to the strength or magnitude of the relationship between two variables. It quantifies the degree to which the variables are related and provides information about the practical significance of the correlation.